# Regulatory Roles of Estrogens in Psoriasis

**DOI:** 10.3390/jcm11164890

**Published:** 2022-08-20

**Authors:** Akimasa Adachi, Tetsuya Honda

**Affiliations:** 1Department of Dermatology, Tokyo Metropolitan Bokutoh Hospital, Tokyo 130-8575, Japan; 2Department of Dermatology, Kyoto University Graduate School of Medicine, Kyoto 606-8507, Japan; 3Department of Dermatology, Hamamatsu University School of Medicine, Hamamatsu 431-3192, Japan

**Keywords:** psoriasis, female sex hormone, estrogen, progesterone

## Abstract

Psoriasis is a common chronic inflammatory skin disease of the interleukin (IL)-23/IL-17 axis. The severity of psoriasis has been reported as higher in men than in women. The immunoregulatory role of female sex hormones has been proposed to be one of the factors responsible for sex differences. Among female sex hormones, estrogens have been suggested to be significantly involved in the development of psoriasis by various epidemiological and in vitro studies. For example, the severity of psoriasis is inversely correlated with serum estrogen levels. In vitro, estrogens suppress the production of psoriasis-related cytokines such as IL-1β and IL-23 from neutrophils and dendritic cells, respectively. Furthermore, a recent study using a mouse psoriasis model indicated the inhibitory role of estrogens in psoriatic dermatitis by suppressing IL-1β production from neutrophils and macrophages. Understanding the role and molecular mechanisms of female sex hormones in psoriasis may lead to better control of the disease.

## 1. Introduction

Psoriasis is a common chronic inflammatory disease with well-demarcated red scaly plaques throughout the body [1]. The prevalence of psoriasis is estimated to be approximately 0.5~8.5% of the worldwide population [2]. Although the pathogenesis of psoriasis has not been fully elucidated, it is now widely accepted that the interleukin (IL)-23/IL-17 axis is a central pathway in psoriasis development, especially in plaque-type psoriasis [3]. In psoriatic lesions, IL-23 is primarily produced by inflammatory dendritic cells (DCs) [1]. IL-23, together with IL-1β, induces IL-17A/F and IL-22 production in various IL-17-producing cells, such as Th17/Tc17 and γδT cells [4,5]. IL-17/22 then activates keratinocytes to produce inflammatory molecules/chemokines such as chemokine (C-X-C motif) ligand (CXCL)-1, 2, and 8; chemokine (C-C motif) ligand 2 (CCL-2); and CCL-20, which recruit inflammatory cells including neutrophils, inflammatory macrophages, and T cells to the skin and accelerate psoriatic inflammation [1,6]. T cells, macrophages, and keratinocytes produce tumor necrosis factor-α (TNF-α) and amplify these cytokine networks [6]. Other than these cytokines, IL-36 and IFN-a are mainly involved in the development of pustular psoriasis and paradoxical psoriasis, respectively [7]. In addition to these central pathways, various genetic and environmental factors are involved in the modification of psoriasis development, and female hormones are suggested to be disease-modifying factors [8].

Estrogens are representative female hormones that are produced mainly in the ovaries. Estrogens play an important role in controlling the female sexual cycle, pregnancy, and childbirth. However, estrogens may also be involved in regulating immune cell functions [9]. For example, estrogen suppresses nuclear factor-κ B (NF-κB) and mitogen-activated protein kinase (MAPK) signaling and downregulates inflammatory responses in various cell populations in vitro [9,10]. However, it remains unclear whether these immune-regulatory functions of estrogens play physiologically significant roles in inflammatory diseases, including psoriasis.

In this short review, we summarize the current findings regarding the involvement of estrogen in the pathogenesis of psoriasis.

## 2. Physiology of Estrogens

### 2.1. Physiological Levels of Estrogens

Estrogens are a group of steroid hormones present in three major physiological forms: estrone (E1; molecular weight (MW) 270.4 g/mol), 17β-estradiol (E2; MW 272.4 g/mol), and estriol (E3; 288.4 g/mol). Estrogens are mainly produced from cholesterol in the ovaries. Estrogens are also produced in the liver, heart, skin, brain, male testes, adrenal glands, and fat tissues [11]. E2 is the most abundant and potent estrogen at the reproductive age. In males, serum E2 levels are less than 40 pg/mL [12], whereas, in females, serum E2 levels range between 30 and 800 pg/mL during the menstrual cycle and increase up to 20,000 pg/mL during pregnancy [11]. After menopause, the serum E2 levels decrease to <20 pg/mL. In the postmenopausal period, serum E2 levels decrease by 85–90% from the mean premenopausal level [12].

### 2.2. Estrogen Receptors and Their Signaling

Estrogen signaling is primarily mediated through two estrogen receptors (ERs)—ERα and ERβ—which are expressed in a wide variety of cell types, including neutrophils, monocytes/macrophages, T cells, and DCs [9]. ERα and ERβ genes are encoded by *Esr1* and *Esr2* and these genes are located on 6 and 14 chromosomes, respectively. E2 binds to these receptors to form dimers, which translocate to the nucleus (Figure 1). In the classical genomic pathway, the dimers bind to estrogen response elements (ERE), and activate the target gene expression. In the non-classical pathway, the dimers interact with other transcription factors, such as NF-κB, specificity protein 1 (SP1), activator protein-1 (AP-1), and CCAAT/enhancer binding protein β (C/EBPβ), and prevent their binding to the transcription factor regulatory elements, leading to the inhibition of their target gene expression [13,14,15]. Of note, these transcriptional factors control the gene expression of many psoriasis-related cytokines and chemokines. For example, NF-κB is involved in the transcription of genes such as IL-23, IL-1β, TNF-α, CCL-2 and CXCL-1; SP1 in IL-1β and TNF-α; and AP-1 and C/EBPβ in IL-23 and IL-36, respectively [16,17,18,19,20].

In addition to these major receptors, G protein-coupled estrogen receptor 1 (GPER1, also known as GPR30), which is located in the endoplasmic reticulum and plasma membrane, binds to E2 with a high affinity [21]. GPER1/GPR30 mediates estrogen signaling through nongenomic responses, including activation of the mitogen-activated protein kinase (MAPK) signaling cascade, cAMP formation, insulin-like growth factor 1 receptor (IGFR), epidermal growth factor receptor (EGFR) and intracellular calcium mobilization [22]. Nuclear ERs mediate signals slowly over hours or days, whereas GPER1 responds much faster, even within seconds [13] (Figure 1).

## 3. Epidemiological and Case Series Studies about the Possible Involvement of Estrogens in Psoriasis

There are various epidemiological studies investigating the prevalence and severity of psoriasis in men and women. Some studies indicate that the prevalence and severity of psoriasis are higher in men than in women [23,24,25,26,27,28,29,30], especially at the estrogen abundant age [31], while other reports failed to observe significant differences in the prevalence of psoriasis between men and women [32,33,34] (Table 1). A recent systematic review indicates that the prevenance is similar between men and women, but the severity in women is lower than men [35]. The age of disease onset is also different between men and women. For example, a German study demonstrates that the age of onset has two peaks, one occurring at the age of 16 years in women or 22 years in men, and a second at the age of 60 years in women or 57 years in men [34]. Recent studies indicate that the two peaks for age at onset are around 18–29 and 50–59 years in women, whereas they are around 30–39 and 60–69 or 70–79 years in men [36]. During pregnancy, in which serum levels of female hormones dramatically change, approximately 33–55% of patients with psoriasis show improvement in symptoms, although some patients, especially patients with pustular psoriasis, occasionally show exacerbated symptoms during pregnancy [37]. In contrast, in the postpartum period, approximately 65% of psoriasis patients exhibit worsening of skin lesions associated with decreased levels of female sex hormones [38,39,40,41,42,43]. Serum levels of E2 and the relative ratio of serum levels of E2 to that of progesterone correlate with psoriasis severity in pregnant patients with psoriasis [38]. Serum E2 levels are inversely correlated with psoriasis severity [44]. Low-dose E2 administration induces improvement in psoriatic arthritis [43], but it is not effective against pustular psoriasis and plaque-type psoriasis [45,46]. On the other hand, it has been reported that tamoxifen, an antiestrogen agent, results in the remission of psoriasis, whose symptoms worsen during a perimenstrual cycle [47]. These studies suggest that estrogens have both proinflammatory and anti-inflammatory roles in psoriasis.

## 4. In Vitro Studies Regarding the Immuno-Regulatory Action of E2

Keratinocytes and various immune cells orchestrate psoriatic inflammation in psoriatic lesions. In this section, we introduce in vitro studies that investigated the potential anti-inflammatory roles of E2 in each cell population (Table 2).

### 4.1. Keratinocytes

Keratinocytes play a critical role in psoriasis development [50,53,60]. Keratinocytes release multiple factors, such as damage-associated molecular patterns (DAMPs), CCL-20, and CXCL-1, 2, and 8 [1]. In vitro, the production of chemokines, such as RANTES and CCL-2, is inhibited by E2 in normal human keratinocytes [51,52]. Isoflavone genistein, which is the major metabolite of soy that binds to human ERα and ERβ, decreases MAPK, signal transducer and activator of transcription 3 (STAT3), NF-κB, and phosphatidylinositol-3 kinase (PI3K) activation in human keratinocytes, leading to decreased mRNA expression of *CCL20*, *S100A7*, and *S100A9* induced by IL-17A and TNF-α [48,49,54]. These results suggest that E2 downregulates keratinocyte activation in psoriatic lesions.

### 4.2. Neutrophils, Monocytes, and Macrophages

Infiltration of neutrophils into the epidermis is one of the characteristic histological findings in psoriasis. Although the actual roles of neutrophils/monocytes/macrophages in psoriasis development are still not fully understood, there are some case studies suggesting disease-promoting roles of neutrophils/monocytes/macrophages in psoriasis. For example, psoriatic lesions have been reported to significantly improve during drug-induced agranulocytosis [55]. Granulocyte and monocyte apheresis therapy ameliorates the symptoms of psoriasis [56]. In mouse studies, depletion of neutrophils has been shown to attenuate psoriasis symptoms [55,56,57,58,59]. These studies suggest that neutrophils, monocytes, and macrophages facilitate the development of psoriasis.

Some studies have investigated the effects of E2 on cytokine production from neutrophils/monocytes/macrophages, but the results are not necessarily consistent among reports. For example, physiological to supraphysiological levels of E2 downregulated TNF-α and IL-1β production from human monocytes and macrophages, whereas no inhibitory effects of physiological to supraphysiological levels of E2 were observed in other studies [9,61,62,63,64]. E2 may exert bidirectional effects on TNF-α and IL-1β production by monocytes and macrophages, depending on its concentration.

In addition to the effects on cytokine production, the inhibitory roles of E2 on neutrophil functions, such as superoxide anion (O_2_^−^) generation, degranulation, and apoptosis, have been reported [65,66].

### 4.3. DCs

DCs play a critical role in the development of psoriasis by producing IL-23 and TNF-α [6]. In vitro, supraphysiological levels of E2 impair IL-23 production from murine bone marrow-derived DCs [67], suggesting that E2 plays a regulatory role in psoriasis development, especially during pregnancy. In contrast, the physiological levels of E2 facilitate IL-1β production in murine vaginal CD11c+DCs [68] and CXCL-8 and CCL-2 production in human monocyte-derived DCs [69], suggesting that the influence of E2 on DC functions differs depending on the concentration and type of DCs.

### 4.4. T cells

T cells (Th17/Tc17) produce pro-inflammatory cytokines such as IL-17A and TNF-α in psoriatic lesions and significantly contribute to psoriatic inflammation [6]. To date, few studies have investigated the role of E2 in Th17/Tc17 cell functions, but some in vitro studies have suggested inhibitory roles of E2 on Th17 cell differentiation and activation [70].

In vitro, physiological levels of E2 inhibited Th17 differentiation through downregulation of retinoid orphan receptor gamma t (Rorγt) expression in murine splenic T cells [70,71,72]. The effect of E2 on IL-17 production in Th17/Tc17 cells has not been investigated, but it has been reported that supraphysiological levels of E2 inhibit TNF-α production in human T cells, suggesting that E2 at high concentrations, such as during pregnancy, may downregulate TNF-α production from Th17 cells in psoriatic lesions. However, it has also been reported that E2, at physiological concentrations, enhances TNF-α production [73,74]. The molecular mechanisms that determine the concentration-dependent effects of E2 on T-cell function remain unclear.

Other than Th17 differentiation, involvement of estrogen on Th1/Th2 differentiation has been reported [75]. For example, physiological levels of E2 inhibited Th1 differentiation through downregulation of T-bet, and shifted toward Th2 differentiation in murine T cells in the lymph nodes and spleen [72,76]. Since Th1-type immune responses play facilitating roles in psoriasis while Th2-type immune responses counterbalance Th17-type immune response [77], estrogens may also play inhibitory roles in psoriasis by down-regulating Th1 and up-regulating Th2 differentiation.

## 5. In Vivo Studies Regarding the Role of E2 on Psoriatic Inflammation

As mentioned above, the possible inhibitory or facilitating roles of E2 in psoriatic inflammation have been suggested by various epidemiological and in vitro studies [78]. However, it remains unclear whether and how E2 plays a role in psoriatic inflammation in vivo. Currently, two in vivo studies have investigated the role of E2 in psoriatic inflammation [79]. Iwano et al. examined the role of E2 in psoriatic inflammation using an imiquimod-induced psoriasis model. Male BALB/c mice were used in the psoriasis model and E2 was administered exogenously. The mice treated with E2 showed exacerbated dermatitis. Administration of an ERα agonist also exacerbated dermatitis. Furthermore, the production of IL-23 by DCs was enhanced by E2 and an ERα agonist in vitro. Based on these data, it was suggested that E2 plays a pro-inflammatory role in psoriasis by inducing IL-23 through ERα [79].

In contrast, we observed anti-psoriatic roles of E2 in the same mouse model [80]. To investigate the role of E2 in psoriatic inflammation in vivo, we applied ovariectomized female C57BL/6 mice, in which the endogenous production of female hormones, including E2, is almost impaired, to an imiquimod-induced psoriasis model. Ovariectomized mice exhibited exacerbated psoriatic inflammation, whereas exogenous administration of E2 reversed the exacerbation, suggesting that E2 plays an anti-psoriatic role physiologically. The anti-psoriatic effects of E2 were mediated through ERα and ERβ in neutrophils and macrophages. Mechanistically, E2 downregulated IL-1β production in neutrophils and macrophages, leading to decreased IL-17A production in γδT cells. The inhibitory effect of E2 on IL-1β production has also been observed in human polymorphonuclear and mononuclear cells. This result may explain the fluctuating IL-1β levels during the female reproductive cycle in humans, in which IL-1β levels are higher during the luteal phase (low serum E2 level) and lower during the follicular phase (high E2 level) [71,81,82,83]. Together, these results suggest that E2 plays a suppressive role in psoriatic inflammation in mice through the regulation of neutrophil and macrophage functions such as IL-1β production. It remains unclear why different effects of E2 were observed in the two studies. E2 may play both pro- and anti-psoriatic roles in a context-dependent manner (Figure 2), as suggested in previous in vitro studies.

## 6. Concluding Remarks

The immunoregulatory mechanisms of E2 in psoriasis, which have mostly been investigated in in vitro studies, have gradually been elucidated in vivo using a mouse psoriasis model. Recognition by patients and clinicians of the potential impact of sex hormones including E2 would lead to a better management of psoriasis symptoms, especially in women. Furthermore, data in the mouse psoriasis model suggest that an appropriate activation of estrogen receptor-signaling is a potential novel therapeutic strategy in psoriasis. However, there are some important issues to be solved before estrogens can be used as a treatment for psoriasis. First, since systemic estrogen therapy has various undesired side effects such as an increased risk of thrombosis and endometrial cancer, and that psoriasis patients tend to develop cardiovascular diseases, topical estrogen therapy, rather than systemic therapy, may be practical. Second, since there are many differences in the pathogenesis between mouse and human psoriasis models, we need to be cautious when applying the findings of mouse studies to human psoriasis. For example, neutrophils and macrophages are the major sources of IL-1β in a murine psoriasis model, while keratinocytes may be the primary source of IL-1β in human psoriasis [5]. Thus, the anti-psoriatic effects of E2 through the inhibition of IL-1β production by neutrophils and macrophages may be limited in human psoriasis. The molecular mechanisms that switch the functions of E2 from pro-inflammatory to anti-inflammatory in psoriasis remain unclear and should be further investigated. Investigation of the involvement of other female hormones such as progesterone in psoriasis is also of interest. In fact, during pregnancy, psoriasis symptoms improve in some patients, whereas they worsen in others, suggesting the existence of female hormones that facilitate psoriasis. In addition, it has been reported that the administration of progesterone flares pustular psoriasis [37,84], suggesting that progesterone may play a facilitating role in psoriasis. Thus, there still remains many unsolved issues on the roles of sex hormones in psoriasis and for the translation of the findings to clinical practice. Nevertheless, elucidation of these issues may lead to the development of novel treatment strategies for psoriasis from the perspective of sex hormones.

## Figures and Tables

**Figure 1 jcm-11-04890-f001:**
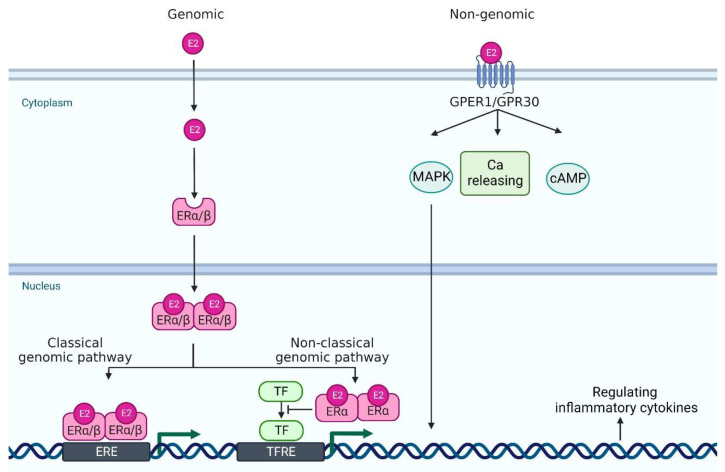
A scheme of estrogen receptors and the intracellular signaling pathway. In genomic pathway, 17β-estradiol (E2) binds to estrogen receptor α and estrogen receptor β in the cytoplasm. It forms dimer and translocates to the nucleus. Then, they bind to estrogen receptor element (ERE) and activate the transcription of downstream genes (classical genomic pathway). Or, they interact with other transcription factor (TF)s, such as NF-κB, specificity protein 1 (SP1), activator protein-1 (AP-1), and CCAAT/enhancer binding protein β (C/EBPβ), and prevent their binding to the transcription factor regulatory element (TFRE) (non-classical genomic pathway), leading to the regulation of their target gene expression. In non-genomic pathway, E2 binds to G protein-coupled estrogen receptor 1 (GPER1) and it regulates mitogen-activated protein kinase (MAPK), calcium (Ca) release, and cyclic adenosine monophosphate (cAMP). Created with Biorender.com.

**Figure 2 jcm-11-04890-f002:**
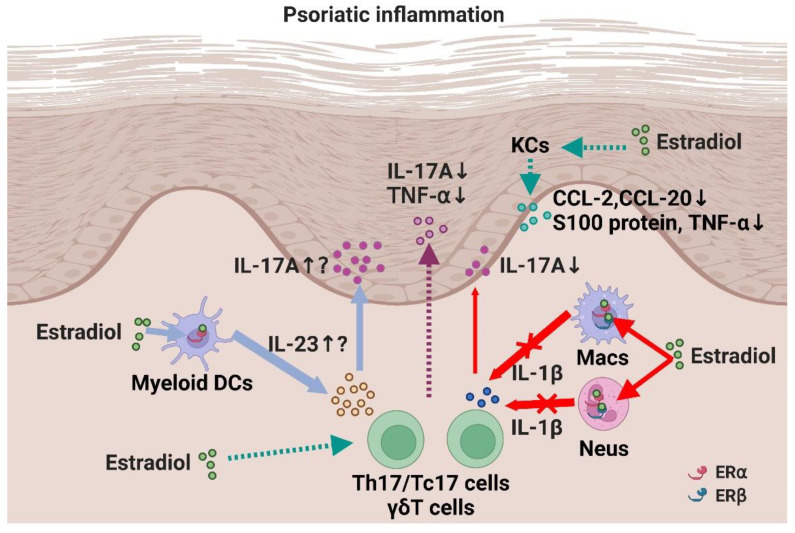
A scheme of possible functions and the mechanisms of estrogen in psoriatic inflammation. E2 play anti-psoriatic functions by downregulating IL-1β production from neutrophils (Neus) and monocytes/macrophages (Macs) through ERα and ERβ. However, in a certain condition, E2 may play facilitating role on psoriatic inflammation by inducing IL-23 production from dendritic cells (DCs) through ERα. Solid lines show the findings from in vivo studies and dotted lines show the findings from in vitro or other disease model studies. Created with Biorender.com.

**Table 1 jcm-11-04890-t001:** A summary of previous reports on the prevalence ratio of psoriasis between men and women.

	Prevalence Ratio of Psoriasis	
	Men	Women
Farber 1974 [32]	46%	54%
Henseler 1985 [34]	50.8%	49.2%
Kawada 2003 [26]	65.80%	34.20%
Takahashi 2009 [25]	66.40%	33.60%
Tsai 2011 [30]	61.60%	38.40%
Furue 2011 [24]	72%	28%
Na 2013 [29]	54.60%	45.40%
Lee 2017 [28]	57.30%	42.70%
Hӓgg 2017 [23]	59.80%	40.20%
Bayaraa 2018 [31]	67.10%	32.9%
El-komy 2020 [27]	56.30%	43.70%
Armstrong 2021 [33]	48.60%	51.40%

**Table 2 jcm-11-04890-t002:** In vitro studies regarding the effects of estrogen on immune cell functions related to psoriatic inflammation.

	Estrogen
Keratinocytes	RANTES↓(physiological to high) [40]
	CCL-2↓(physiological to high) [39]
	CCL-20↓(isoflavone) [42]
	S100A7↓(isoflavone) [42]
	S100A9↓(isoflavone) [42]
Neutrophils	superoxide anion (O_2_^−^)↓(not mentioned) [48]
	degranulation↓(high) [48]
	apoptosis(physiological to high) [49]
	migration↓(physiological to high) [49]
Monocytes/Macrophages	IL-1β→~↓(high) [50,51,52]
	TNF-α→~↓(high) [51,52,53]
Dendritic cells	IL-23↓(high) [54]
	IL-1β↑(physiological) [55]
	IL-8↑(high) [56]
	CCL-2↑(high) [56]
T cells	IL-17↓(physiological) [57]
	TNF-α↓(high), TNF-α↑(low) [58,59]

RANTES, Regulated on activation, normal T cell expressed and secreted; CCL, CC-chemokine ligand; S100A7, S100 calcium-binding protein A7; S100A9, S100 calcium-binding protein A9; O_2_^−^, superoxide anion; IL, interleukin; TNF, tumor necrosis factor.

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
