# Peer review of "Regulatory Roles of Estrogens in Psoriasis"

_jcm, 2022, doi:10.3390/jcm11164890_

Round 1

Reviewer 1 Report

The authors have presented an interesting review article about regulatory roles of estrogens in psoriasis. The overall scope is well met. However, there are a few significant concerns that can be appropriately addressed to improve the paper. My remarks are the following: 

  1. The first part - physiology of estrogens is definitely to short and should include more information about the impact of estrogens on mechanism,  which are connected with pathogenesis of psoriasis.

2. Line 130, 142 - there are no references (should be add in the end of the sentences)

3. Concluding remarks are too short and too general. It could have more details about possible impact of estrogens on psoriasis. 

Author Response

Reviewer#1

The authors have presented an interesting review article about regulatory roles of estrogens in psoriasis. The overall scope is well met. However, there are a few significant concerns that can be appropriately addressed to improve the paper. My remarks are the following: 

  • The first part - physiology of estrogens is definitely too short and should include more information about the impact of estrogens on mechanism,which are connected with pathogenesis of psoriasis.

We appreciate the reviewer’s insightful comments. Following the comments, we added information about the impact of estrogen on molecular mechanisms which are connected with pathogenesis of psoriasis as below.

 “These transcriptional factors control the gene expression of many psoriasis-related cytokines and chemokines. For example, NF-κB is involved in the transcription of genes such as IL-23, IL-1β, TNF-α, CCL-2 and CXCL-1, SP1 in IL-1β and TNF-α, and AP-1 and C/EBPβ in IL-23 and IL-36, respectively [15–19].” (page 6, line 82-85)

We also added information about the involvement GPER1-signaling on the activation of IFGR and EGFR-signaling, both of which are important for the epidermal hyperplasia in psoriasis as below.

“GPER1/ GPR30 mediates estrogen signaling through nongenomic responses, including activation of the mitogen-activated protein kinase (MAPK) signaling cascade, cAMP formation, insulin-like growth factor 1 receptor (IGFR), epidermal growth factor receptor (EGFR) and intracellular calcium mobilization [21]” (page 6, line 90-92)

  • Line 130, 142 - there are no references (should be add in the end of the sentences)

We apologize for our mistakes. We added the references (reference68 and 76).

  • Concluding remarks are too short and too general. It could have more details about possible impact of estrogens on psoriasis.

We appreciate the important comment. Following the reviewer’s suggestion, we expanded the discussion by adding the descriptions about the possible impact of estrogen on psoriasis as well as the potential translational approach of estrogen as a therapeutic option in psoriasis as below.

“Recognition by patients and clinicians of the potential impact of sex hormones including E2 would lead to a better management of psoriasis symptoms, especially in women. Furthermore, data in mouse psoriasis model suggest that an appropriate activation of estrogen receptor-signaling is a potential novel therapeutic strategy in psoriasis. However, there are some important issues to be solved before estrogens can be used as a treatment for psoriasis. First, since systemic estrogen therapy has various undesired side effects such as an increased risk of thrombosis and endometrial cancer, and that psoriasis patients tend to develop cardiovascular diseases, topical estrogen therapy, rather than systemic therapy, may be practical.” (page 14-15, line 230-238)

and

“Thus, there still remains many unsolved issues on the roles of sex hormones in psoriasis and for the translation of the findings to clinical practice. Nevertheless, elucidation of these issues may lead to the development of novel treatment strategies for psoriasis from the perspective of sex hormones.” (page 15, line 252-255)

Reviewer 2 Report

Dear Sir/ma'am,

Few basic facts have to be corrected:

Males and females are equally affected by psoriasis vulgaris. Many studies indicate that age of onset is younger in females. Thus, one German study demonstrated a peak age of onset of 22 years in males and 16 years in females in early-onset disease. 

Ref: Henseler T, Christophers E. Psoriasis of early and late onset: characterization of two types of psoriasis vulgaris. J Am Acad Dermatol 1985; 13: 450–6.

Rook's Textbook of Dermatology

Second: Certain forms of psoriasis worsens in pregnancy like pustular psoriasis.

Third: Estrogen appears to up-regulate Th2 cytokines and down-regulate Th1 and Th17 cytokines.

This point along with down regulation of TNF alpha with estrogen needs o be highlighted.

Also labeling psoriasis and Th17 predominant disease in one line is misleading as the pathogenesis is multifactorial

Also how does this study translate to clinical practice is not clear.

Author Response

Reviewer#2

Few basic facts have to be corrected:

  1. Males and females are equally affected by psoriasis vulgaris. Many studies indicate that age of onset is younger in females. Thus, one German study demonstrated a peak age of onset of 22 years in males and 16 years in females in early-onset disease. 

Ref: Henseler T, Christophers E. Psoriasis of early and late onset: characterization of two types of psoriasis vulgaris. J Am Acad Dermatol 1985; 13: 450–6.

Rook's Textbook of Dermatology

We appreciate the important comment and introducing the references. We agree with the reviewer that there are studies which did not see significant differences in the prevalence of psoriasis between men and women (reference 30-32 in the main text, which are also shown in table 1), although the severity of psoriasis in women is lower than men, which is indicated by a recent systematic review (Guillet et al. International Journal of Women’s Dermatology, 2022).

Following the reviewer’s suggestion, we deleted the sentences describing the male predominance of the prevalence of psoriasis throughout the manuscript. We also added the information about the age of onset in women and men as below. 

 “A recent systematic review indicates that the prevenance is similar between men and women, but the severity in women is lower than men [33]. The age of disease onset is also different between men and women. For example, a German study demonstrates that the age of onset has two peaks, one occurring at the age of 16 years in women or 22 years in men, and a second at the age of 60 years in women or 57 years in men [32]. Recent studies indicate that the two peaks for age at onset are around 18-29 and 50-59 years in women, whereas they are around 30-39 and 60-69 or 70-79 years in men [34].” (page 7, line 102-109)

Second: Certain forms of psoriasis worsens in pregnancy like pustular psoriasis.

We appreciate the reviewer’s comment. We added a sentence describing the worsening of pustular psoriasis in pregnancy as below.

“although some patients, especially patients with pustular psoriasis, occasionally show exacerbated symptoms during pregnancy [35].” (page 7, line 111-112)

Third: Estrogen appears to up-regulate Th2 cytokines and down-regulate Th1 and Th17 cytokines.

This point along with down regulation of TNF alpha with estrogen needs o be highlighted.

We appreciate the reviewer’s comment. Following the reviewer’s suggestion, we expanded the descriptions (section 3.4.) about the effects of estrogen on Th1/Th2 differentiation as below. 

“Other than Th17 differentiation, involvement of estrogen on Th1/Th2 differentiation has been reported [73]. For example, physiological levels of E2 inhibited Th1 differentiation through downregulation of T-bet, and shifted toward Th2 differentiation in murine T cells in the lymph nodes and spleen [70,74]. Since Th1-type immune responses play facilitating roles in psoriasis while Th2-type immune responses counterbalance Th17-type immune response [75], estrogens may also play inhibitory roles in psoriasis by down-regulating Th1 and up-regulating Th2 differentiation.” (page 11-12, line 186-192)

Also labeling psoriasis and Th17 predominant disease in one line is misleading as the pathogenesis is multifactorial

Following the reviewer’s suggestion, we explained the involvement of cytokines other than IL-17 in the development of psoriasis as bellow.

“Other than these cytokines, IL-36 and IFN-a are mainly involved in the development of pustular psoriasis and paradoxical psoriasis, respectively [7].” (page 3, line 44-46)

Also how does this study translate to clinical practice is not clear.

We appreciate the important comment. At present, there still remains many unsolved issues on the roles of sex hormones in psoriasis, and thus it is still difficult to translate the findings in basic researches to clinical practice. However, recognition by patients and clinicians of the potential impact of sex hormones on psoriasis symptoms may lead to a better management of psoriasis. In addition, further elucidation of the functions of sex hormones in psoriasis may lead to a novel treatment option in psoriasis. To make these points clear, we added descriptions in the concluding remark part as below.

“Recognition by patients and clinicians of the potential impact of sex hormones including E2 would lead to a better management of psoriasis symptoms, especially in women. Furthermore, data in mouse psoriasis model suggest that an appropriate activation of estrogen receptor-signaling is a potential novel therapeutic strategy in psoriasis. However, there are some important issues to be solved before estrogens can be used as a treatment for psoriasis. First, since systemic estrogen therapy has various undesired side effects such as an increased risk of thrombosis and endometrial cancer, and that psoriasis patients tend to develop cardiovascular diseases, topical estrogen therapy, rather than systemic therapy, may be practical.” (page 14-15, line 230-238)

and

“Thus, there still remains many unsolved issues on the roles of sex hormones in psoriasis and for the translation of the findings to clinical practice. Nevertheless, elucidation of these issues may lead to the development of novel treatment strategies for psoriasis from the perspective of sex hormones.” (page 15, line 252-255)

Round 2

Reviewer 1 Report

No further comments

Author Response

-

Reviewer 2 Report

the comments are addressed and the study can be accepted

Author Response

-